# Meta-Analysis on Intervention Effects of Physical Activities on Children and Adolescents with Autism

**DOI:** 10.3390/ijerph17061950

**Published:** 2020-03-17

**Authors:** Jinfeng Huang, Chunjie Du, Jianjin Liu, Guangxin Tan

**Affiliations:** 1School of Physical Education, South China Normal University, Guangzhou 510631, China; 2019020923@m.scnu.edu.cn; 2College of Vocational and Technical Education, South China Normal University, Nanhai 528000, China; duchunjie@m.scnu.edu.cn; 3Department of Physical Education, Guangzhou College of Commerce, Guangzhou 511363, China

**Keywords:** physical activities, autism, children, adolescents, meta-analysis

## Abstract

This paper aimed to discuss the intervention effects of physical activities on children and adolescents with autism with a meta-analysis so as to serve as a reference to further relevant research on the same topic. As for research methods, by searching in CNKI (China National Knowledge Infrastructure), WanFang data, VIP Database for Chinese Technical Periodicals, PubMed, Scopus, Web of Science, and other databases, this study collected randomized controlled trials (RCTs) on the intervention of physical activities on children and adolescents with autism and used Review Manager 5.3 software to process and analyze the outcome indicators of the literature. As for the result, a total of 12 papers and 492 research targets were selected. The results of the meta-analysis show that physical activity had a significant positive impact on social interaction ability, communication ability, motor skills, and autism degree of autistic children as well as the social skills and communication skills of autistic adolescents. On the other hand, physical activity had no significant effect on the stereotyped behavior of autistic children and adolescents. In conclusion, physical activity intervention is beneficial to children and adolescents with autism, and continuous physical activity intervention can produce greater intervention effect.

## 1. Introduction

Autism is a common developmental disorder characterized by difficulties with speech and behaviors, such as lack of social abilities, repetitive behaviors, communication disorders, activity and interest disorders with limitations [1]. The disease mainly occurs in children and adolescents, but the explicit age limits of children and adolescents haven’t been decided. Based on the degree of individual cognition and socialization, Zhang Wenxin divides the kids under 12 years old into children and kids between 13 and 22 years old into adolescence [2]. According to the latest data from the Centers for Disease Control and Prevention, the prevalence rate of autistic children is 1/59. And the data from Asia, Europe and North America reports an average prevalence of 1–2% [3]. According to the report on the development of Chinese autistic children released by China on 17 October 2014, the prevalence of autism in China is similar to that in other countries in the world, with about 1%. It can be seen that autism has become a public health problem that seriously affects the health of children and adolescents.

In addition to the core symptoms of social interaction, communication, and stereotyped behavior, the motor skills of autistic patients are usually stunted [4]. At present, many researches on the intervention of physical activity on children and adolescents with autism have been conducted. Physical activity refers to all kinds of physical actions that consume energy due to skeletal muscle contraction, including physical exercise, work, housework, entertainment and other activities [5]. However, due to social and behavioral defects, autistic patients usually show a decline in physical activity level. Fewer opportunities for physical activity are more likely to affect their behavior [6], and cause some chronic diseases, such as obesity, which is very common in autistic patients [7]. It has been reported that physical activity has a good intervention effect on autistic patients [8], and the physical and mental health of autistic patients contains the improvement of core symptoms [9]. Some studies have shown that after the intervention of physical activity, all of the social interaction ability [10,11,12], communication ability [11,13,14,15], stereotyped behavior [11,16] and sports skills [11,17,18,19] of children and adolescents with autism have been improved, which can reduce the degree of autism [14,15,20,21]. These studies show that physical activity is very effective in intervention of children and adolescents with autism. 

Although a large number of studies have shown that physical activity is effective in the intervention of children and adolescents with autism, the results of individual studies may still be uncertain due to the great differences in sample size, intervention means, intervention time, intervention frequency and measurement results in various studies. Therefore, this meta-analysis was carried out on studies of the intervention of physical activity on children and adolescents with autism of the in the latest 10 years to objectively evaluate the intervention effect of physical activity on social interaction ability, communication ability, stereotyped behavior, sports skills and autism degree of children and adolescents with autism so as to provide the basis for clinical application of physical activity intervention in the operation of children and adolescents with autism.

## 2. Materials and Methods

### 2.1. Selection and Exclusion Criteria of Literature

Literature including all of these criteria were selected: (1) the research type was experimental, particularly randomized control experiments; (2) the research target were children and adolescents with autism; (3) the test group had obvious physical activity intervention, while the control group did not carry out any physical activities; (4) the outcome indicators were either autism rating scales or motor skills rating scale. Literature including any of these criteria were excluded: (1) they were non-randomized controlled trials; (2) subjects were neither children nor adolescents with autism; (3) they are repetitive published literature; (4) intervention means were non-physical activities; (5) the literature results showed no outcome index data; (6) they were reviews; (7) the control group had physical activity intervention.

### 2.2. Searching Strategies

Literature searching was conducted in CNKI, WanFang Data, VIP Database for Chinese Technical Periodicals, PubMed, Scopus, Web of Science, and other databases with “free words + subject words”, including Chinese and English journals, conference papers, and dissertations. Key words both in English and Chinese, such as physical activities, physical exercise, sports, and autism, were used. The literature searching period was from January 2010 to December 2019.

### 2.3. Data Extraction and Quality Evaluation

#### 2.3.1. Data Extraction and Processing

Data on authors, publication time, physical activity contents, test groups, and control groups (sample sizes, measurement tools for autistic patients, means and standard deviations before and after intervention) were extracted. When different scales were used in each experiment, standardized mean, difference, SMD (standardized mean difference), and 95% confidence intervals (CIs) were selected as the combined statistics. When the same scales were used in each experiment, weighted mean difference (WMD) and 95% CIs were selected as the combined statistics. If there was heterogeneity in the results, the random effect model was used to analyze the data; if not, the fixed effect model was used. I^2^ represents the heterogeneity of various studies [22].

#### 2.3.2. Literature Quality Evaluation

We evaluated the quality of the literature according to the Jadad scale [23]. The evaluation included three aspects: random and its hidden program, blinding, withdrawal, and loss to follow up. The scale was from 0–5; a 0–2 score was considered low-quality research, and a 3–5 score was considered high-quality research.

## 3. Results

### 3.1. General Results of Selected Research Literature

Figure 1 depicts the detailed literature search process. A total of 1357 primary literature were obtained by using literature searching strategy. After the selection of reading topics and abstracts, 240 papers were analyzed. In the second round, the following conditions were excluded: (1) the intervention means were not physical activities; (2) the research type was not an RCT experimental study; (3) there was no outcome index data; (4) the control group was physical activity; (5) they were reviews; (6) subjects were not children nor adolescents with autism. Finally, 12 articles were included in the meta-analysis.

### 3.2. General Features of Selected Research Literature

#### 3.2.1. General Information of Each Study

Through comprehensive search of relevant literature and strict selection according to the inclusion and exclusion criteria, 12 randomized controlled trials finally met the inclusion criteria for systematic evaluation (see Table 1). All of the 12 papers were published within the past decade. A total of 492 children and adolescents with autism were included in 12 articles, including 253 in the experimental group and 239 in the control group.

#### 3.2.2. Interventions of Physical Activities

The relevant literature included in this study all used physical activities as intervention means, including sports games, water sports, football, aerobics, karate, horse riding, and other physical activities.

#### 3.2.3. Frequency of Physical Activity Intervention

In seven papers [10,12,13,14,15,16,20], the frequency of physical activity intervention was more than 3 times per week; in four papers [17,18,19,21], the frequency of physical activity intervention was 3 times or less; and in one paper [11], the frequency of intervention was not reported.

#### 3.2.4. Duration of Each Physical Activity Intervention

Among 12 selected papers, 11 [10,11,12,13,14,15,16,17,18,19,21] mentioned the duration of each physical activity intervention (between 45 and 90 min).

#### 3.2.5. Physical Activity Intervention Cycle

The intervention cycle of physical activity ranged from 4 weeks to 24 weeks. There were 5 papers [11,15,18,19,21] with an intervention period less than 12 weeks (including 12 weeks), and 7 papers [10,12,13,14,16,17,20] with an intervention period more than 12 weeks.

### 3.3. Results

#### 3.3.1. Quality Evaluation of Selected Literature

Among the 12 papers included in meta-analysis, four [14,15,20,21] were Chinese and eight [10,11,12,13,16,17,18,19] were English. All papers had a clear autism behavior score scale or motor skills score scale. Among the 12 literatures, 11 [10,11,12,13,14,15,16,17,18,19,20] mentioned randomness. Two of them [11,19] mentioned randomization and described the correct randomization method. None of the 12 papers mentioned the implementation of a blind method, and no one quit the intervention process. According to the Jadad quality rating scale, these papers were evaluated from three aspects: random, blinding, loss to follow up or withdrawal. Nine [10,12,13,14,15,16,17,18,20] were scored 1, 1 [21] was scored 0, and 2 [11,19] were scored 2. In this case, papers selected by this method had poor qualities.

#### 3.3.2. Impacts of Physical Activities on Social Interaction Ability of Autistic Children and Adolescents

In a meta-analysis of three studies (*n* = 197), physical activity significantly improved the social interaction ability of autistic children and adolescents (SMD = −0.58, 95% CI: −0.87 to −0.29, I^2^ = 3%, z = 3.95, *p* < 0.0001, Figure 2).

#### 3.3.3. Impacts of Physical Activities on Communication Ability of Autistic Children and Adolescents

In a meta-analysis of 4 studies (*n* = 240), physical activity significantly improved communication ability in children and adolescents with autism (SMD = −0.29, 95% CI: −0.55 to −0.04, I^2^ = 35%, z = 2.25, *p* = 0.02 < 0.05, Figure 3).

#### 3.3.4. Impacts of Physical Activities on Stereotyped Behaviors of Autistic Children and Adolescents

Among the 12 papers included, two [11,16] (*n* = 146) compared the scores of stereotyped behaviors between the control group and the experimental group. Through the heterogeneity test (Chi^2^ = 1.82, df = 1 (*p* = 0.18), I^2^ = 45%). Since *p* > 0.1, I^2^ < 50%, according to Cochrane Manual, heterogeneity is acceptable, and therefore the fixed effect model was used for analysis. The results showed that the horizontal line and the diamond, after combining the two studies, intersected with the invalid line, indicating that the combination statistics of multiple studies were not statistically significant (SMD = −0.13, 95% CI: −0.46 to 0.20, I^2^ = 45%, z = 0.78, *p* = 0.43 > 0.05, Figure 4). That is, the influence of physical activities on the stereotyped behaviors of children and adolescents with autism was not significant.

#### 3.3.5. Impacts of Physical Activities on Motor Skills of Autistic Children and Adolescents

In the four selected papers [11,17,18,19] (*n* = 172), the total sample size of the experimental group was 86, and the total sample size of the control group was 86. There was heterogeneity among the four research results (I^2^ = 89%); thus, the random effect model was used for analysis. The results showed that the combined statistics of multiple studies were not statistically significant (SMD = −0.17, 95% CI: −1.46 to 1.11, I^2^ = 89%, z = 0.26, *p* = 0.79 > 0.05, Figure 5).

The data in Figure 5 show high heterogeneity (I^2^ = 89% > 50%). Therefore, sensitivity analysis is needed to determine the heterogeneity sources of physical activities on motor skills of autistic children and adolescents. Firstly, we removed Reference [19] with a biggest difference from others but found differences still existed. Then, the method of reducing one study at a time was used to test whether each study had a significant impact on the combined effect of motor skills of autistic children and adolescents. The sensitivity analysis showed that the heterogeneity dropped to 0 after deleting References [11,19] which had a significant impact on the results of the concomitant effect. Another analysis of References [11,19] showed that the research objects of study [11] were autistic children and adolescents aged from 6 to16, while those of References [17,18] were children aged from 4 to 12. Therefore, the difference may be caused by different research objects. Reference [19] had a shorter intervention period (6 weeks). To sum up, long-term physical activity intervention may not have a significant impact on the motor skills of autistic adolescents but has a significant impact on the motor skills of autistic children. In the meta-analysis of two studies (*n* = 38), physical activity significantly improved motor skills of autistic children (SMD = 1.02, 95% CI: 0.33 to 1.71, I^2^ = 0%, z = 2.91, *p* = 0.004 < 0.05, Figure 6).

#### 3.3.6. Impacts of Physical Activities on Levels of Autism of Children and Adolescents

A meta-analysis was conducted on 4 papers [14,15,20,21] on the level of autism in children and adolescents. According to the results of heterogeneity test (chi2 = 19.75, df = 3 (*p* = 0.0002), I^2^ = 85% (I^2^ > 50%) which indicated that there was significant heterogeneity in the four literatures. Therefore, the random effect model was adopted for analysis, and the results showed that the combined statistics of multiple studies had no statistical significance (SMD = −1.14, 95% CI: −2.02 to −0.25, z = 2.52, *p* = 0.01 < 0.05, Figure 7).

The data in Figure 7 show high heterogeneity (I^2^ = 85% > 50%). Therefore, in order to find the sources of heterogeneity, sensitivity analysis should be carried out, and the method of reducing one document at a time should be adopted to check whether each document has a significant impact on the results of merger effect. Sensitivity analysis shows: The deletion of Jiang Feng’s [21] study had a significant effect on the results of merger effect. Another analysis on Jiang Feng [21] showed that the subjects of this study were autistic adolescents aged from 12 to18 years old. The intervention frequency of physical activity was three times a week. Its difference from the other three studies [14,15,20] is the age of the subjects and the intervention frequency of physical activity. So, the difference may be caused by the age of the subjects or the intervention frequency of physical activity. In the meta-analysis of the three studies (*n* = 154), physical activity significantly decreases the autism level in autistic children (SMD = −0.61, 95% CI: −0.94 to −0.29, I^2^ = 0%, z = 3.72, *p* = 0.0002 < 0.05, Figure 8).

## 4. Discussion

This paper is a systematic review of quantitative data analysis on the physical activity effects of children and adolescents with autism. All eligible randomized controlled trials were published between 2010 and 2019. The results showed that physical activity can significantly improve the social interaction ability and communication ability of children and adolescents with autism, improve the motor skills, and reduce the degree of autism of children with autism.

The social and communication defects in individuals with autism make them less likely to participate in sports activities and get along with people [6], which may lead to the sedentary lifestyle of autistic patients. Not only does it affect the overall health of autistic patients, but it also further deprives them of social adaptability. Studies have shown that physical activity is beneficial to the health of autistic patients [24]. Through organized physical activities, people with autism have the opportunity to communicate with others. In the six [10,11,12,13,14,15] articles about the influence of physical activity on the social interaction ability and communication ability of children and adolescents with autism, all the researchers mentioned the way of guiding the patients to play games. Autistic children have a lower quality of games experience than non-autistic children of the same intellectual age [25]. Perhaps for children and adolescents with autism, these physical activities are games they are playing. In the case that children cannot use language to express completely and clearly, playing games is a good communication method [26].

Stereotyped behavior is one of the core defects of autistic patients, which is a series of repetitive, aimless and meaningless behaviors [27]. The physical activity on the reduction of stereotyped behavior of autistic patients may be explained by the fact that the stimulation obtained by physical activities of autistic patients has similar internal mechanism of action with the stimulation produced by stereotyped behavior, which can bring comfortable sensory stimulation to autistic patients, so that they can achieve appropriate level of excitation through sensory stimulation and adjustment [28]. However, the results of this review show that physical activity has no significant effect on the stereotyped behavior of children and adolescents with autism. Through a careful analysis of the two articles [11,16], it can be seen that Reference [16] refers to the intervention of physical activity combined with conventional treatment on children and adolescents with autism, while Reference [11] does not. Therefore, the statistical difference may be caused by different interventions. Physical activity, a single intervention, may be combined with conventional therapy to intervene in children and adolescents with autism, and the effects of these two interventions on children and adolescents with autism should be considered in future studies.

Heterogeneity of results has been found in the analysis of four articles [11,17,18,19] on the influence of physical activity on motor skills of autistic children and adolescents, but it disappears when two articles [11,19] are excluded. Through the analysis of two articles with significant effects [17,18], the intervention cycle of physical activity is 12–32 weeks, with each intervention lasting 45–75 min and 1–2 times per week. Furthermore, the subjects of intervention are all children with autism aged from 4 to 12. It can be concluded that physical activity intervention can improve the sports skills of children with autism, while exerting little effect on the sports skills of adolescents with autism.

Eventually, among the effects of physical activity on the degree of autism of children and adolescents, the positive effect of physical activity on the degree of autism of children is significant, excluding the article [21] of the research object including adolescents. Therefore, physical activity can be considered effective in reducing the degree of autism in children, but it may have little effect on the degree of autism in adolescents.

According to the results of this meta-analysis, physical activity in the cycle of 4–24 weeks, 4–13 times a week, 30–90 min each time is conducive to the improvement of social interaction ability and communication ability in children and adolescents with autism, and physical activity in the cycle of 4–24 weeks, 5–7 times a week, 90 min each time is conducive to the decrease of degree of autism in children. However, the improvement of sports skills of children by physical activities requires long-term intervention, at least 12 weeks, 45–75 min each time, 1–2 times a week. In conclusion, continuous physical activity intervention is beneficial to children and adolescents with autism.

## 5. Limitations of Current Research

There are some limitations and shortcomings in this meta-analysis. First and foremost, there are relatively few RCTS on physical activity intervention in individuals with autism, and consequently the number of articles included was small. Furthermore, most of the random methods in the study were not clear, and all the articles do not mention the allocation concealment and blinding, which may have a certain negative impact on the evaluation results of the system. Last but not least, among the 12 articles included, 4 of them [14,16,18,20] have mentioned that physical activity combined with routine rehabilitation treatment is used to intervene the patients, while the rest 8 [10,11,12,13,15,17,19,21] do not describe whether there is routine rehabilitation treatment for the patients during the intervention period, while routine treatment during the intervention period may affect the intervention effect of physical activity on children and adolescents with autism. Therefore, in order to confirm the positive effect of physical activity children and adolescents with autism, physical activity can be adopted alone to intervene patients in the future research.

## 6. Conclusions

The results of this meta-analysis show that after physical activity intervention, the social interaction ability and communication ability of children and adolescents with autism have been improved, and the motor skills of children with autism have been improved as well, while the degree of autism of children has been reduced. In conclusion, physical activity intervention is beneficial to children and adolescents with autism, and continuous physical activity intervention can produce greater intervention effect. In addition, in the future research, children and adolescents with autism can be intervened with other physical activities, which may also be a reference for improving other symptoms of children and adolescents with autism through physical activities in the future.

## Figures and Tables

**Figure 1 ijerph-17-01950-f001:**
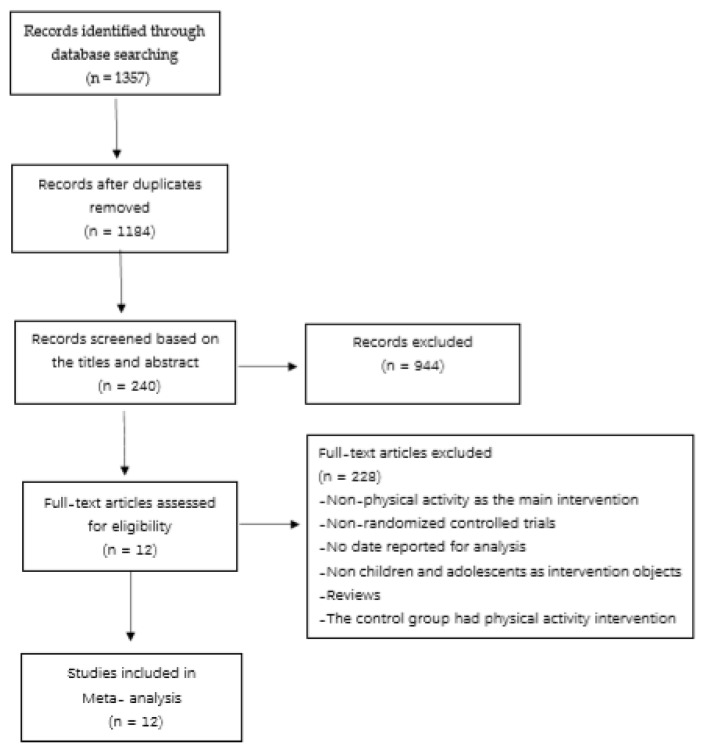
Literature selection process.

**Figure 2 ijerph-17-01950-f002:**
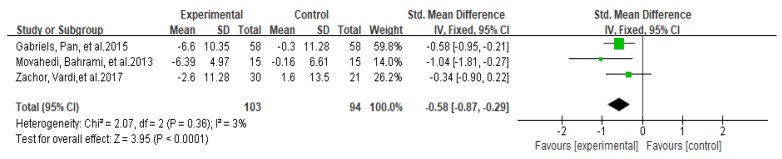
Forest plot of meta-analysis on impacts of physical activities on social interaction ability of autistic children and adolescents.

**Figure 3 ijerph-17-01950-f003:**
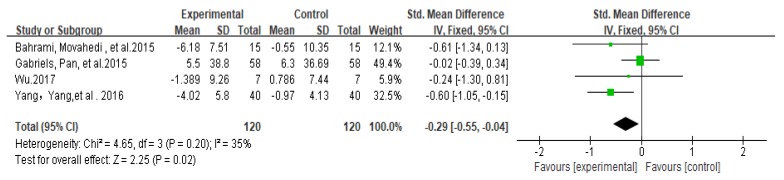
Forest plot of meta-analysis on impacts of physical activities on communication ability of autistic children and adolescents.

**Figure 4 ijerph-17-01950-f004:**
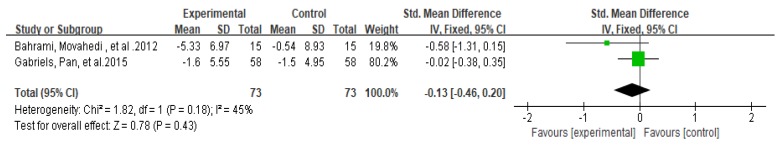
Forest plot of meta-analysis on impacts of physical activities on stereotyped behaviors of autistic children and adolescents.

**Figure 5 ijerph-17-01950-f005:**
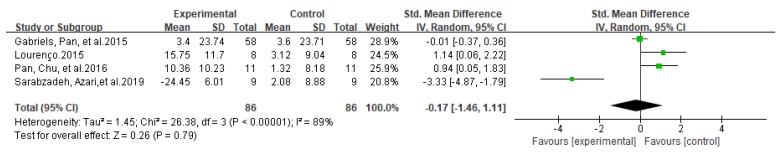
Forest plot of meta-analysis on impacts of physical activities on motor skills of autistic children and adolescents.

**Figure 6 ijerph-17-01950-f006:**
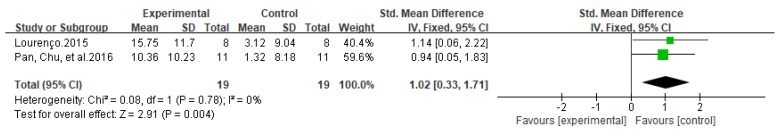
Forest plot of meta-analysis on impacts of physical activities on motor skills of autistic children.

**Figure 7 ijerph-17-01950-f007:**
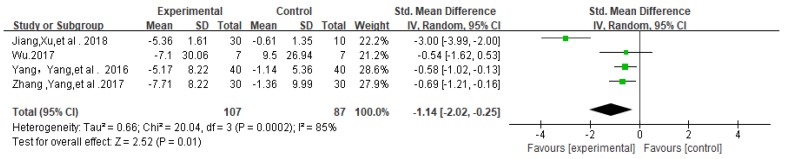
Forest plot of meta-analysis on impacts of physical activities on levels of autism of children and adolescents.

**Figure 8 ijerph-17-01950-f008:**
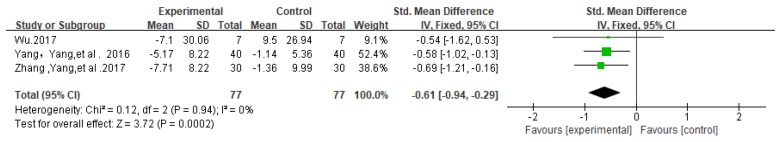
Forest plot of meta-analysis on impacts of physical activities on levels of autism of children.

**Table 1 ijerph-17-01950-t001:** General features of selected research literature.

Study	Participants		Intervention Program	Physical Activity Intervention			Measuring Tools
	Sample Size (*n*)	Age (years)		Frequency (weekly)	Time (min)	Duration (week)	
Movahedi, Bahrami, et al. (2013) [10]	30	5–16	EG: Kata techniques	4	30 (week 1 to week 8)	14	Social Communication subscale of GARS-2
			CG: No exercise intervention		90 (week 9 to week 14)		
Gabriels, Pan, et al. (2015) [11]	116	6–16	EG: Therapeutic horseback intervention	NR	45	10	BOT-2; PPVT-4; SALT; SRS; VABS-II
			CG: Completed a brief artistic project				
Zachor, Vardi, et al. (2017) [12]	51	3–7	EG: Outdoor activities	13	30	13	SRS
			CG: No exercise intervention				
Bahrami, Movahedi, et al. (2015) [13]	30	5–16	EG: Karate training	4	30 (week 1 to week 8)	14	Communication subscale of GARS-2
			CG: No exercise intervention		90 (week 9 to week 14)		
Yang, Yang, et al. (2016) [14]	80	3–10	EG: Sports game	6	90	24	CARS Scale; ATEC
			CG: Regular education				
Wu (2017) [15]	14	4–10	EG: Water sports training	5	90	4	ABC
			CG: No exercise intervention				
Bahrami, Movahedi, et al. (2012) [16]	30	5–16	EG: Karate training	4	30 (week 1 to week 8)	14	Stereotypy subscale of GARS-2
			CG: No exercise intervention		90 (week 9 to week 14)		
Lourenço (2015) [17]	16	4–10	EG: Trampoline training	6	90	24	BOT-2
			CG: No exercise intervention				
Pan, Chu, et al. (2016) [18]	22	6–12	EG: Table tennis	2	70	12	BOT-2; WCST
			CG: No intervention				
Sarabzadeh, Azari, et al. (2019) [19]	18	6–12	EG: Tai Chi	3	60	6	MABC-2
			CG: No exercise intervention				
Zhang, Yang, et al. (2017) [20]	60	3–11	EG: Sports game	7	NR	24	CARS Scale; CBS
			CG: No exercise intervention				
Jiang, Xu, et al. (2018) [21]	25	12–18	EG: Sports activities	3	62	12	ABC
			CG: No exercise intervention				

EG = experimental group; CG = control group; CARS Scale = Child Autism Rating Scale; ATEC = Autism Treatment Evaluation Checklist; CBS = Clancy Behavior Scale; ABC = autism behavior checklist; GARS-2 = Gilliam Autism Rating Scale-Second Edition; BOT-2 = Bruininks–Oseretsky Test, 2nd Ed; PPVT-4 = Peabody Picture Vocabulary Test, 4th Ed; SALT = Systematic Analysis of Language Transcripts; SRS = Social Responsiveness Scale; VABS-II = Vineland Adaptive Behavior Scales-II; WCST= Wisconsin Card Sorting Test; BASC = Behavior Assessment System for Children; MABC-2 = Movement Assessment Battery for Children-Second Edition; NR = not reported.

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
