# Peer review of "Meta-Analysis on Intervention Effects of Physical Activities on Children and Adolescents with Autism"

_ijerph, 2020, doi:10.3390/ijerph17061950_

Round 1

Reviewer 1 Report

The authors took into account my comments and they proceeded with the necessary revisions. I thank the authors their time and effort. I do believe that the manuscript has improved considerably.

Author Response

Dear Reviewer :
Thank you for your reviews on our manuscript " Meta Analysis on Intervention Effects of Physical Activities on Children and Adolescents with Autism". Thoses reviews are very helpful for us to revise and imporve our paper. We used the "revision" feature in Microsoft Word to make the changes. Please see the attachment.The main corrections in this paper are as follows:

English style:

  • We have revised the English style of the abstract, introduction, discussion and conclusion.

Abstract:

  • We removed digital information from article abstracts.
  • In lines 21 to 26, we used more direct sentences.

Results:

  • The definition and quality of figure 1 have been improved.
  • In Table 2, just below the sub-header “Sample size” insert (n). The same with “Age”, insert just below (years).
  • In line 69, we changed the number 86 to 89.

The above is what we modified.

Thank you and best regards.
Yours sincerely,
Huang Jinfeng

Reviewer 2 Report

Physical activity has positive effects on: 1) social interaction, 2) communication ability, 3) improved motor skills, and 4) the degree of autism was significantly. On the other hand, physical activity has no influence on: 1) stereotyped behavior

MAJOR CONCERNS:

I am not a native English, but from my point of view the reading is quite harsh. I suggest a thorough revision of the English style and the utilization of more direct sentences.

MINOR CONCERNS:

Abstract:

  1. It is not necessary to include so much statistical and numerical information in the abstract. Only the main messages. The statistical information will be present in the results section.
  2. The abstract is messy. You have 5 results from your meta-analysis. It could be nice to cite first the positive ones and after the no-significant influence (referring to the stereotyped behavior.
  3. Basically you should write a more direct sentence like: physical activity did not influenced/affected/ changed/…. Stereotyped behavior
  4. In line 30 you wrote. “the degree of autism of children in the physical activity group was significantly improved”. Improve has a meaning similar to increase (I think that you wanted to say exactly the opposite). I suggest writing that the degree of autism diminished significantly. In order to be coherent with the message in lines 62-63. “which can reduce the degree of autism [14,15,20,21]
  5. In line 56, I propose a more direct sentence of the kind: Less opportunities for physical activity affect negatively their behavior.

Results:

  1. Definition and quality of the Figure 1 should be improved
  2. In Table 2, just below the sub-header “Sample size” insert (n), doing that way you don’t have to insert in each row “n= …”. The same with “Age”, insert just below (years). In fact I suggest you the same formal strategy used by yourself in the same table in the sub-headers “Frequency”, “Time” and “duration”.

Author Response

Dear Reviewer :
Thank you for your reviews on our manuscript " Meta Analysis on Intervention Effects of Physical Activities on Children and Adolescents with Autism". Thoses reviews are very helpful for us to revise and imporve our paper. We have studied these reviews carefully and tried our best to revise and imporve the manuscript. We used the "revision" feature in Microsoft Word to make the changes. Please see the attachment.The main corrections in this paper are as follows:

English style:

1、We have revised the English style of the abstract, introduction, discussion and conclusion.

Abstract:

  • We removed digital information from article abstracts.
  • In lines 21 to 26, we used more direct sentences.

Results:

  • The definition and quality of figure 1 have been improved.
  • In Table 2, just below the sub-header “Sample size” insert (n). The same with “Age”, insert just below (years).
  • In line 69, we changed the number 86 to 89.

The above is what we modified.

Thank you and best regards.
Yours sincerely,
Huang Jinfeng

This manuscript is a resubmission of an earlier submission. The following is a list of the peer review reports and author responses from that submission.

Round 1

Reviewer 1 Report

Dear authors:

There are some concerns from this reviewer about your manuscript, of which I present some of the most relevant:

English language writed expressions should improve so that understanding mostly at the method key points. Why have you not used the PRISMA method, which is commonly accepted internationally? The absence of the Web of Science and Scopus databases among those selected in the first screening is very upseting, why is the rationale behind this omission? The inclusion criteria of the first selected publications should be clarified In the case of being this study a standard meta analysis, shouldn't the effect sizes have been calculated and presented? As the authors surely know, such strict criteria set aside studies that do not meet randomization, excluding a lot of studies, because it is very difficult to carry out this kind of studies in small or very heterogeneous samples. Why have not included the BMI among the variables to consider, according to the characteristics of patients with ADS, whis is whitout any doubt very relevant to the practice of PA? Regarding the latter, the great diversity of symptoms, typologies and characteristics of the patients diagnosed under the ADS umbrella makes it necessary to include some scale among the variables to be considered in the included studies. Has the medication (e.g., stimulants, antipsychotics) prescribed to these patients been taken into account? This data is present in some of the selected studies, as it can interfere with the practice of PA. Finally, there should be a double list of bibliographic references: 1) the studies on which the theoretical framework and design are based, and 2) that of the selected studies, in order to analyze if there are some biases of any sign.

Reviewer 2 Report

In relation to the manuscript entitled "Meta Analysis on Intervention Effects of Physical 3 Activities on Children and Adolescents with Autism", indicate that it can be accepted after making minor changes:

It is recommended that authors improve the introduction of the article. For example, they can deepen the expected benefits of the practice of PA in autism or other related elements. It would also be of interest if they clearly stated the objective at the end of the introduction

The flow chart should appear before, in the method section.

Throughout the manuscript the expression “Error! Reference source not found ”. This must be remedied.

Adapt references according to the regulations of the magazine (they have errata).

Reviewer 3 Report

Thank you for the opportunity to review the manuscript entitled, "Meta Analysis on Intervention Effects of Physical Activities on Children and Adolescents with Autism”.

I believe this study investigated a topic relevant to the readers of “IJERPH”.  There is a growing interest in research into the determinants of health-related quality of life. Studies highlighting the importance of regular physical activity as a healthy practice and numerous investigations have shown that the practice of physical activity contributes positively to psychosocial wellbeing in the population. Systematic reviews (SRs) and meta-analyses (MAs) are considered as the best tools to synthesize the scientific evidence as to which treatments, interventions, or prevention programs should be applied for a given problem.

This paper is well written and follows well accepted standards of academic writing. However, major revisions may prove beneficial. In this current form the manuscript needs to be improved.

The search of the scientific literature should be expanded to include others databases:  Web of Science, Scopus, SportDiscus,  Science Direct… I thing that a meta-analyses with very few included studies is less informative.

The meta-analysis should provide answers to: What intervention of physical activities is most effective?, What frequency of physical activity is most effective?, What duration of each physical activity is most effective?...

There is no relationship between the results and the discussion. The discussion must be rewritten. The manuscript lacks a clear discussion. 

The figures must be clearly ans easier presented.

In the manuscritp, the word “Stereotypes” must be changed to “Stereotypic behaviors”